# Human Impact on the Twenty-Four-Hour Patterns of Steller Sea Lions’ Use of a Haulout in Hokkaido, Japan

**DOI:** 10.3390/ani14091312

**Published:** 2024-04-27

**Authors:** Yuko Chayahara, Yumiko Nakanowataru, Sara Abe, Runa Kurosawa, Sayuki Suma, Nana Murasato, Rin Oyamada, Natsuki Ebashi, Masatoshi Tsunokawa, Mayu Sakurama, Takanori Kooriyama

**Affiliations:** 1Department of Veterinary Science, Rakuno Gakuen University, 582 Bunkyodai-Midori, Ebetsu 069-8501, Hokkaido, Japan; 2Otaru Aquarium, 3-303 Shukutsu, Otaru City 047-0047, Hokkaido, Japan

**Keywords:** conflict, human disturbance, haulout, herring, Steller sea lions

## Abstract

**Simple Summary:**

The Steller sea lion (SSL) is the largest Otariidae family species widely distributed in the cold North Pacific Ocean. SSLs also migrate to Hokkaido, Japan, in the winter. SSLs travel south along the Sea of Japan coast for winter to catch the herring that spawn near the coast. SSLs utilize haulouts as rest sites during herring season, but this can be affected by weather- and human-related pressures. In this study, we investigated the haulout use patterns of SSLs and the effect of human disturbance over a period of three years. The behavior of the SSLs on the haulouts differed from when they were on rookeries. Furthermore, the number of SSLs on the haulout and the duration of time they spent there reduced from the first year (lower human pressure) to the third year (human presence increased). Based on these findings, the SSLs may adjust the timing of when they enter the water to around sunrise, following the ecology of the herring and to avoid human disturbance. Therefore, it is necessary to continue investigating the relationship between the ecology of the SSLs and environmental changes in order to determine the damage that the SSLs cause to fishery interests.

**Abstract:**

Steller sea lions (SSLs) migrate to the Hokkaido coast to spend the winter there, leading to conflicts arising with fishermen over herring. This study analyzed the trends in the SSLs’ use of a haulout as a rest site under human pressure. From January to March in 2017, 2018, and 2019, we recorded the SSL behavior at the haulout site off Otaru City, Hokkaido, for 24 h a day using a fixed-point video recorder. We investigated three years of data to analyze the relationships between the SSL behaviors (attendance/landing–entry timings/remaining on land) and herring caught. We also monitored the SSL behaviors during changes in weather conditions and under human pressure. Throughout the three years, the SSLs used the haulout site during harsher weather or under human pressure. In 2017 and 2018, there was a correlation between the herring caught and the maximum number of SSLs on the haulout, but not in 2019. The number of SSLs on the haulout increased from evening to night; most individuals entered the water in the morning. The SSLs probably return to the water around sunrise not only for foraging but also to avoid anthropogenic pressure. The damage caused to the herring fishery by the SSLs was severe, but it is also clear that human pressure changed their behavior in response.

## 1. Introduction

Marine mammals have long been vital to human life and are used for food, oil, and fur [1]. However, with the establishment of the Convention for the Protection of Mammals and the development of modern fisheries, marine mammals and humans have become competitors for the same fish stocks [2]. In addition, with the decline in fishery resources, the damage caused by marine mammals to fisheries, especially in coastal fishing, has become significant. In response, humans have chosen to keep marine mammals away from fishing grounds by threatening them with guns and sometimes culling them [3]. In Japan, a few people hunt wild animals for food or raw materials for products [4]. However, in Hokkaido, Japan, where pinnipeds migrate every winter, conflicts with the fishing industry have become a problem, just as in other regions. In Japan, four species of pinnipeds pose problems as fishery damage: Steller sea lions, spotted seals, harbor seals, and fur seals [5]. International animal protection treaties were established, such as the “Convention on International Trade in Endangered Species of Wild Fauna and Flora” and the “Convention on the Conservation of Migratory Species of Wild Animals” in the 1970s. However, it was not until 1993 that marine mammals became the subject of protection in Japan. Two species of seals are now under the jurisdiction of the Ministry of the Environment, while sea lions and fur seals are under the jurisdiction of the Ministry of Agriculture, Forestry, and Fisheries. Fur seals are now protected under the “Sea Otters and Fur Seals Hunting Control Act”, they are no longer allowed to be captured or culled. The two species of seals are protected, but culling has been authorized to control their populations to prevent fishery damage. The hunting of SSLs for meat and lethal control purposes is permitted within limits.

The largest species of the sea lion family, the Steller sea lion (SSL), *Eumetopias jubatus*, is in the middle of the food chain, being a predator for a variety of fish and cephalopods but also a target for its natural enemy, the killer whale [1]. They have a characteristic half-land, half-sea lifestyle and are widely distributed on the cold North Pacific coast, where they live in groups. Among these, the Asian stock mainly give birth and breed in rookeries (breeding grounds) in the Kuril Islands, Sakhalin, and the northern Sea of Okhotsk from June to September. It is not uncommon for pinnipeds to migrate thousands of kilometers. In the case of the SSLs, the migratory period for winter foraging runs from October after the breeding season ends to May of the following year. Thousands of SSLs travel to the coast of Hokkaido to avoid the drift ice of the Sea of Okhotsk, resting at specific haulouts such as reefs and small islands in a few locations, and conserve energy for the next breeding season as they overwinter. [6].

SSLs are an essential source of protein for Alaska Natives, who hunt around 200 SSLs each year. Currently, SSLs are protected in the United States and Russia. Historically, in Japan, approximately 1000 head of SSLs were captured annually, mainly in the southern part of the Kuril Islands, due to a lack of leather during the Sino-Japanese War that began in 1937 [6]. During World War II, approximately 20,000 sea lions were captured using dynamite in the southern Kuril Islands [6]. These days, SSL leather is no longer used for leather products, and only their meat circulates in the market on a small scale. Meanwhile, the amount of herring caught off the coast of Hokkaido, which had continued since the Meiji era, rapidly decreased. After 1954, the herring catch was vastly reduced. At the same time, the fishery damage caused by SSLs increased. Since 1959, the lethal controls of SSLs have been carried out almost without limit, mainly using hunting guns under the support of the national and Hokkaido governmental subsidies. Ohtaishi and Wada [6] noted that 22,481 SSLs were culled in Japan from 1961 to 1992. Loughlin et al. [7] reported that the SSL populations in Far East Russia decreased from 50,000 to 10,000 between 1960 and 1989. The SSL population was categorized as endangered (EN) in the IUCN (International Union for the Conservation of Nature and Natural Resources) red list in the 1990s. Ohtaishi and Wada [5] state that this population reduction might be induced by the reduction in fishery stock, bycatch, and culling in Japan. Still, due to their declining numbers and growing international interest in wildlife protection, the upper limit on the number of SSLs culled per year was changed to 116 from unrestricted in 1994 [6,8]. Afterward, the number of caught SSLs was calculated, considering the balance between damage reduction and conservation, and it was decided that a maximum of 501 SSLs would be culled per year after carrying over the previous year’s unused quota [9]. Other countermeasures have been attempted, such as fireworks, artificial killer whale calls, and string-reinforced gillnets, but the damage continues [5]. Currently, the issue is finding a compromise between species conservation and damage to fisheries. Over the past ten years, the damage caused by SSLs has ranged from JPY 500 million to JPY 2 billion across Japan, at a considerable cost to herring fishery during winter. People in fishery consider them harmful animals in Japan due to issues such as predation, injury, and damage to fishing gear in fisheries [8,10]. The public opinion is concerned with fishery damage rather than marine mammal conservation. One of the reasons for this might be that SSLs have been given a bad image, such as describing them as a “Sea Gang” by the press. In this way, the SSL management issue has been discussed in an unbalanced arena. The management of the SSL conservation in Japan consists of controlling the SSL population, which lacks the assessment of the effect of the lethal control on the SSL ecology. The SSL behavior and its ecological changes must be assessed while the government adopts lethal control to prevent the SSL population reduction by over-culling.

Several thousand SSLs visit Hokkaido every year, and there are around five locations where the groups of nearly 100 SSLs each land [11]. At the haulout “Todo Iwa”, off the coast of Otaru City in Ishikari Bay, Hokkaido, more than 100 SSLs have started to attend, although only a few did so before 2015. It is hypothesized that climate change has weakened drift ice development, which allows the SSLs to breed on Tyuleny Island in Sakhalin [12]. This island is close to the Japanese and Russian channels, and therefore increased numbers of the SSLs flow into the coast of Japan [13,14]. The “Todo Iwa” haulout is located right in front of the Otaru Aquarium and is suitable for observation [11]. This haulout is also close to a nearby port, and therefore hunting boats can reach it quickly. However, an experienced hunter had recently retired, leading to a period of low shooting pressure from 2016 to 2017. In the following years, inexperienced hunters improved their shooting skills, and the human influence gradually increased. Therefore, the behavior of the SSLs during haulout has changed under human pressure. The SSLs remember being hunted when people shot rifles from boats approaching the haulout.

Based on the above background, the purpose of this study was to clarify the human impact on the herds of the wild SSLs that visit the Hokkaido coast and use the haulout site. Concurrently, monitoring the behavior of SSLs will not only demonstrate their ecology at migration haulout sites but also help discover the biological value of the SSLs, which can provide helpful information for resolving human–SSL conflicts.

## 2. Materials and Methods

### 2.1. Animals and Observation Methods

In this study, we observed the flocks of the Asian wild SSLs (*Eumetopias jubatus*) that came to the coast of the Sea of Japan in winter when they landed at a haulout rock (commonly known as Todo Iwa) with a circumference of approximately 400 m located off the coast of Shukutsu, Otaru City, Hokkaido (43°14′31″ N, 141°0′30″ E). This rock is located slightly west of the center of Ishikari Bay, and herring fishing is carried out to the east and west of it. To record the SSLs, we set up a fixed-point video camera (SIGMA 300 mm telephoto lens + Color SONY 960H CCD) at Otaru Aquarium, approximately 200 m from the rock, and recorded footage for 24 h per day.

### 2.2. Hourly Changes in the Number of SSLs, Weather Conditions, and Anthropogenic Impacts

The survey was conducted from 1 January to 31 March in each year from 2017 to 2019. From our video footage, we recorded the number of SSLs that landed using the time sampling of one minute every hour and summarized the daily movements and changes in the maximum number of SSLs that landed. In addition, if an SSL entered the water, we analyzed the video and recorded any human-caused reasons. In classifying the causes, we classified weather and sea conditions as natural factors that force the SSLs to enter the water, and non-lethal scaring boats and escaping from shooting as human factors. To clarify the relationship between these factors, we graphically demonstrated the exact figures for hourly changes in the number of SSLs landing, weather conditions, sea conditions, boat traffic, and shooting.

### 2.3. Changes in the Amount of Prey Resources

SSLs primarily forage on herring in the winter, so changes in herring resources affect SSL behavior. The time period during which the SSLs use a haulout is related to the herring caught near the haulout. Fishery catch information near the haulout was obtained from the catch reports of Otaru City, Ishikari City, and Yoichi Town [15]. In addition, to compare herring catch and SSL haulout use, we divided the monthly data into the early, middle, and late periods of each month, depending on the category of each report.

### 2.4. Analysis Method for Changes in SSL Haulout Usage

#### 2.4.1. Comparison of Landing Start Time and Water Entry Start Time

We investigated when the SSLs started landing on the haulout and when they began entering the water in each of the three years analyzed. The SSLs land when they want to rest, and when they enter the water, they forage. Humans and environmental factors influence the timing of the landing and entry into the water. Human influences include vessel passage and shooting. In particular, hunters shot the SSLs from early morning to midday as they would be able to see the target in these conditions. Therefore, if the SSLs are trying to escape from shooting, they will likely shift their landing times to when it gets dark. Additionally, they become wary of the boats and enter the water early when the boats pass near the haulout.

#### 2.4.2. Comparison of the Residence Time of Haulout per Day

SSLs sometimes remain on a haulout for relatively long periods. By comparing the residence time on the haulout per day over the three years analyzed, we will clarify whether the SSLs’ resting time changed due to other influences. If the SSLs learn through repeated shooting each season, either the number of SSLs using the haulout and the residence time on the haulout will decrease.

### 2.5. Statistical Analysis

Spearman’s rank correlation coefficient was used to determine the correlation between the changes in the herring catch (the early, middle, and late periods of each month) and the number of SSLs landed (the sum of the maximum number of SSLs per day) for each season. We used the Kruskal–Wallis test to compare the distribution of the SSLs’ landing start time and water entry start time over the three years. Furthermore, we also used the Kruskal–Wallis test to see if there was a difference in the residence time spent on the haulout over the three years. *p* < 0.05 was considered to indicate a significant difference. The statistical software R ver. 3.6.1. (The R Foundation, Vienna, Austria) was used for all analyses.

## 3. Results

### 3.1. Visualization of Hourly Changes in the Number of Landed SSLs and Natural and Anthropogenic Influences

The hourly changes in the number of landed SSLs are shown in Appendix A (2017), Appendix A (2018), and Appendix A (2019). The SSLs began to gather in the evening and at night, and their numbers remained high until early morning, but in the morning, many individuals entered the water, so the number of SSLs tended to decrease during the day. Additionally, waves and weather significantly affected the number of SSLs on land; during snowstorms or high waves that swallow rocks, very few SSLs landed, or only a few SSLs stayed for a short time. Even on days when the waves were high, some of them found a small, safe space on the haulout. Furthermore, when boats or people passed in front of the haulout or gunfire was heard, all the SSLs woke up at once and fled to the sea while vocalizing.

From January to March 2017, the days on which no SSLs were observed were 15 and 16 January and 27 and 29 March. The number of SSLs seen varied daily, with a maximum of 105 on 5 February. The SSLs entered the water due to human factors such as the approach of boats. The SSLs entered the water 38 times due to the approaching boats and 3 times after being surprised by people approaching them in kayaks. The lethal control of the SSLs was conducted seven times at the haulout starting in late February. Even after the control efforts began, the SSLs came ashore, but their numbers decreased in March.

From January to March 2018, no SSLs were observed after 6 March. The number of SSLs seen varied daily, with the highest number being 101 on 20 January. The SSLs entered the water nine times due to approaching boats and once when they were startled by a person approaching them in a kayak. The culling of the SSLs began in early January and was conducted a total of 13 times.

From January to March 2019, no SSLs were observed on 18 January; 6–8 and 25 February; and 7, 10, 12, 15, and 18 March and onwards. The number of SSLs seen varied daily, with a maximum of 52 seen on 30 January. The SSLs were observed entering the water five times due to the approach of boats. The culling of the SSLs began in early January and was conducted eight times in total. In 2019, the probability of being hit by a bullet increased, and more individuals were retrieved by boats after dying. Unlike other years, the SSLs did not come ashore in significant numbers after late February. Even after lethal control began, the SSLs still came ashore. From late February onwards, the number of SSL landings decreased dramatically.

### 3.2. Herring Catches and the Number of Landed SSL

According to a 2019 database search of the Fisheries Research Headquarters of the Hokkaido Research Organization [16], the coast near this haulout (encompassing Otaru City, Ishikari City, and Yoichi Town) experienced total herring catches of 722 tons (January–March 2017), 1143 tons (January–March 2018), and 848 tons (January–March 2019). The fishing peaks occurred in late February 2017, early February 2018, and mid-February 2019 [14].

On the other hand, the number of SSLs was counted every hour from January to March in 2017, 2018, and 2019, with the peak occurring in late January (Figure 1). The maximum number of SSLs per day peaked in late January in 2017 and 2019, but in 2018, it peaked in mid-January. Although the number of SSLs was lower in January 2017 than in 2018, the peak in late January was the highest in the three years analyzed, and this maximum number remained until late February and began to decrease in March. The maximum number of SSLs in 2018 per day was higher than in any other year. However, it fell in February and the SSLs were almost absent in March. In 2019, the number of SSLs was the lowest in the three years analyzed, and after the peak in late January, the number rapidly decreased; from late February onwards, they were rarely seen.

Since the SSLs come for herring, their numbers might increase as the herring school spawns. We divided the period from January to March each year by season. We investigated the correlation between the total fish caught and the maximum SSLs that landed per day (Table 1). As a result, there was a strong correlation between the 2017 herring catch and the total maximum number of SSLs that landed per day, with a correlation coefficient of 0.719 (*p* = 0.04). A strong correlation was also found between the 2018 herring catch and the maximum number of SSLs that landed daily, with a correlation coefficient of 0.731 (*p* = 0.04). There was a weak correlation between the 2019 herring catch and the maximum number of SSLs that landed per day, with a correlation coefficient of -0.286, but there was no significant difference (*p* = 0.5).

### 3.3. Comparison of Hauling out Time and Water Entry Start Time at Haulout over Three Years

As shown in Figure 2, the SSLs hauled out from 7:00 to 17:00. No differences were seen in the three years analyzed (*p* = 0.454). Next, as shown in Figure 3, the water entrance start times were mainly concentrated in the morning, with a peak observed between 5:00 and 6:00. In 2017, the SSLs frequently entered the water at 23:00. However, when compared over the three years analyzed, no significant difference was observed (*p* = 0.266). Additionally, entry into the water did not occur between 7:00 and 18:00.

### 3.4. Comparison of SSL Haulout Residence Time over Three Years

We tallied the time during which the SSLs continuously remained on the haulout, divided it into 6 h intervals, and created a histogram based on the number of days (Figure 4). As a result, the residence time on the haulout peaked at 7–12 h in 2017, but the longest time was 104 h on 3–4 March. In 2018, the peak time was 13–18 h, and the maximum residence time was 94 h from 11 to 14 February, a slight decrease compared to the previous year. In 2019, the peak time was 13–18 h, similar to the previous year, but the maximum residence time was 48 h from 5 to 7 January, less than half of that. Each year, there were days when the usage time exceeded 24 h, but the maximum length of the stay decreased from 2017 to 2019. However, no significant difference in the length of the stay on the haulout over the three years was found (*p* = 0.0874, Kruskal–Wallis test).

From 2017 to 2019, the peak length remained unchanged, but the maximum residence time decreased.

## 4. Discussion

### 4.1. Hourly Fluctuations in the Number of SSLs Landed and the Effects of Natural and Anthropogenic Influences

It is reported that the duration of the use and the number of pinnipeds on the haulouts are affected by local prey abundance [17], tidal height, the time of day [18,19], human disturbance, and tidal stage [18]. The pinnipeds on haulouts are also highly sensitive to boats, kayaks, and other human disturbances and quickly enter the water in response [20]. The number of SSLs landed was greatly influenced by boats, people, weather, and waves. The SSLs tend to be selective of the haulouts and must avoid being battered by water, even when faced with serious storms [21]. Therefore, the SSLs did not land on the haulout during heavy snowstorms or high waves strong enough to overwash it. In the case of boats, they were not concerned about boats passing behind the haulout (in a direction they could not see directly), regardless of the size or distance. However, they were extremely wary of boats passing in front of the haulout (directly visible) or approaching boats. Whenever such a boat passed, most SSLs immediately dived into the water. This suggests that the SSLs remember the instances of being hunted when people shot rifles from boats approaching from the front of the haulout.

### 4.2. Changes in Herring Catch and Number of Landed SSLs

Pacific herring inhabit northeast Asian and North American waters, from the coasts of Japan, Korea, and China to Russia, and from Alaska and the Bering Sea to Baja, California [22]. They are renowned for spawning in shallow water, which turns white and cloudy and can be seen from winter to spring. Herring fishing along the Sea of Japan coast of Hokkaido began in the Edo period and continued in large quantities from the Meiji period to the early Showa period [23]. However, after reaching a maximum of 970,000 tons in 1897, the number continued to decline; by 1954, there was almost nothing to catch. This abundance of herring was from the Hokkaido/Sakhalin stock and was seen from March to May. The current main source of herring is the Ishikari Bay stock, the fishing season for which is from January to March. During the period when the abundance of the Hokkaido/Sakhalin stock was high, there was little conflict between fishermen and SSLs [5]. However, whether the reduction was due to environmental changes or overfishing is still being determined. However, as mentioned above, the number of herring caught has drastically decreased [22]. The Ishikari Bay stock became the main source of herring in place of the Hokkaido/Sakhalin stock, but the number of herring caught from the former stock is smaller than the latter. During periods when the stocks of herring were low, conflicts with SSLs destroying the nets became an issue, and countermeasures against SSLs are currently being taken [5]. The release of herring larvae began in 1996, but it is challenging to speculate if this project has been successful. The stock of herring in the Ishikari Bay has increased, and the herring catch in Hokkaido has been expected to reach 20,000 tons by 2022. The increase in herring catch indicates resource improvement [24].

This study showed that the maximum number of SSLs that landed during haulout in 2017 and 2018 was related to the amount of herring caught. Therefore, the SSLs migrated to this area to prey on the herring that gathered there to spawn. The number of SSLs that landed in 2019 was the lowest compared to 2017 and 2018, and the groups did not land from late February onwards. In March 2019, when the SSLs disappeared from the haulout, there was a report that about 70 SSLs had landed on a concrete breakwater about 20 km away [25]. The preference for haulouts depends on prey abundance [26] and shelter from wind or waves [26] in pinnipeds. In other words, SSLs that desert a haulout might find another safer haulout without leaving a herring-rich area. Since it was still the herring fishing season, the SSLs continued eating prey in this new area.

### 4.3. Changes in Landing Start Time and Water Entry Start Time at the Haulout

Most pinnipeds, including SSLs, are catholic feeders and are not selective about migratory fish, root fish, and cephalopods. They eat whatever is abundant in the area and easy to catch [10]. Even if they deviate somewhat from their original daily activity, they adopt a flexible feeding pattern based on the distribution of prey and sea conditions [27]. SSLs often fish at night in Russian waters around rookeries [28,29,30]. Herring is the main prey of the SSLs around the haulouts in the Sea of Japan. The best time to fish is at sunrise when the herring is more visible under the sunlight [31,32]. Based on our results, we did not see the SSLs land between 18:00 and 6:00, and we can deduce that they tried to rest before this time, in other words, at night-time. In addition, considering that the water entrance start time was around 5:00 to 6:00 am, before sunrise, the SSLs intended to forage while the sun was rising. When pinnipeds land at haulouts, it might be related to the amount of sunlight entering the sea and prey availability [18,19]. However, Kucey [33] noted that tidal cycles had a higher correlation with SSL hauling-out behavior according to results from eight locations. This was also related to the fact that the tides caused the submersion of the haulout, reducing the number of spaces to land [34]. The haulout in this study was virtually unaffected by the tides. Therefore, it was suggested that prey availability and other factors influence behavior more. Marine mammals, such as dolphins, have been proven to have high cognitive abilities, and sea lions have also been demonstrated to have high learning and memory abilities [35,36]. When fishermen retrieve their nets, sea lions steal the fish, so the fishermen try to drive them away by making loud noises from the speaker and using bear bunkers (explosion sound). However, although these methods had some effect, the sea lions quickly learned and resumed stealing fish. In 2018 and 2019, the culling frequency increased from January onwards and began after sunrise. Therefore, the SSLs may have returned to the water before the hunters came. In 2017, there was a peak of water entry around 23:00, but there was the possibility that the SSLs dived at night to catch prey caught in the fishermen’s nets.

### 4.4. Changes in SSL Haulout Usage Time

Although they are marine mammals, SSLs differ from cetaceans in that they must come ashore at times during their lifetime [31]. In rookeries, male SSLs fast while maintaining their territory, and even females continue to nurse for several days while fasting [1]. Adult male SSLs store fat to survive the cold winter, and the stored nutrients are an essential source of nutrition for the breeding season [37,38]. According to the results, the maximum time that the SSLs remained on the haulout was two days or more, and they may spend even longer hours on land. In 2018 and 2019, when human pressure increased compared to 2017, there was a tendency for the SSLs to spend less time on the haulout. Although there was no statistically significant difference, it was clear that the maximum rest time decreased in 2019. The time when the SSLs landed was the time when they were able to rest. It is also conceivable that the fishermen can fish in peace while the SSLs rest after foraging. One solution may be keeping the SSLs on the haulout while the fishermen fish for herring.

## 5. Conclusions

In this study, we investigated the trends of the SSLs on a haulout and the impact of human pressure in an area where large numbers of SSLs have begun to land from 2016 onwards. Data over the past ten years (Appendix A) show that fishery damages caused by the SSLs in Ishikari Bay have remained almost unchanged [39]. Therefore, it is unlikely that controlling the SSLs has successfully reduced damage to herring fishing, but some fishermen have positive opinions on it [5]. Public opinion also has a bad image of SSLs; people such as researchers, aquariums, and ecotourists who should advocate for the protection of SSLs find it difficult to object to lethal control. In the 1990s, the sea lion population drastically decreased in the United States and Russia, and it was designated an endangered species and protected. The differences in the responses of the two countries from Japan may reflect that both countries rely less on coastal fishing and have a higher public awareness of wildlife protection [6]. Herring is a fish species that can live around 10–20 years; the abundance of herring will increase or decrease depending on the effects of environmental changes [40]. If there were a way to resolve this conflict, it would be to restore the ocean’s abundance of resources. Therefore, a large total catch may encourage the fishermen to overlook the damage caused by the SSLs. These days, some studies show the importance of marine mammal feces to coastal nutrients. Several whale species turn the fish they feed into excrement and release it into the ocean as a new source of nutrition for fish and plankton [41]. Therefore, the fixation of nutrients on the coast through the migration of marine mammals is extremely important for marine biodiversity. For these reasons, researchers must first demonstrate to the fishermen that short-term measures centered on culling are meaningless for SSL management. Additionally, they need to show that marine mammals may be helping the increasingly abundant herring colonize the Hokkaido coast by providing nutrients to the coast.

## Figures and Tables

**Figure 1 animals-14-01312-f001:**
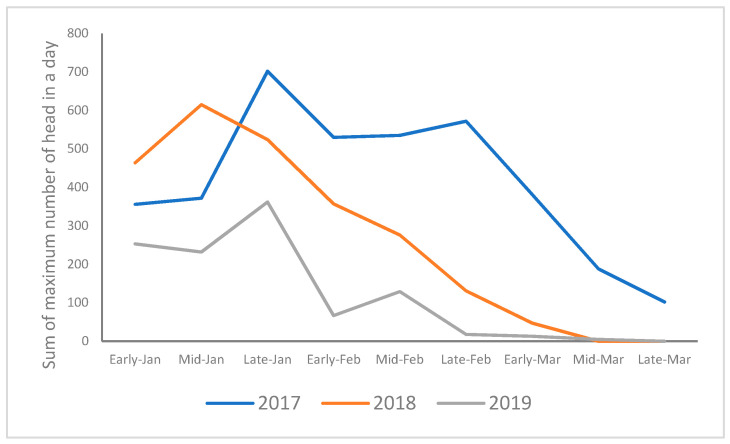
The total number of SSLs on the haulout each part of the month.

**Figure 2 animals-14-01312-f002:**
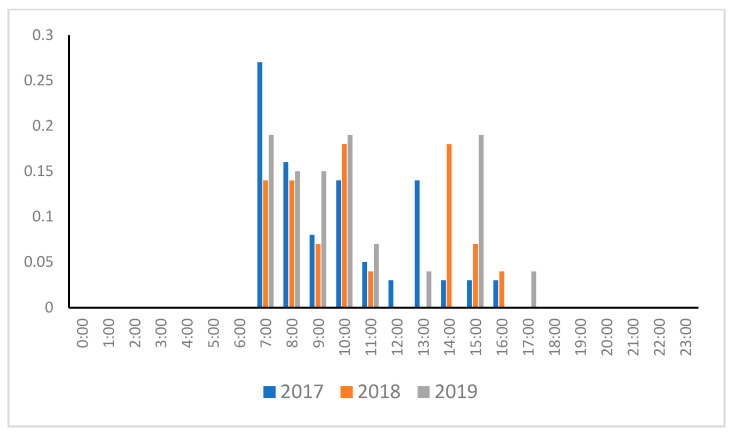
Hauling out start time from January to March 2017–2019.

**Figure 3 animals-14-01312-f003:**
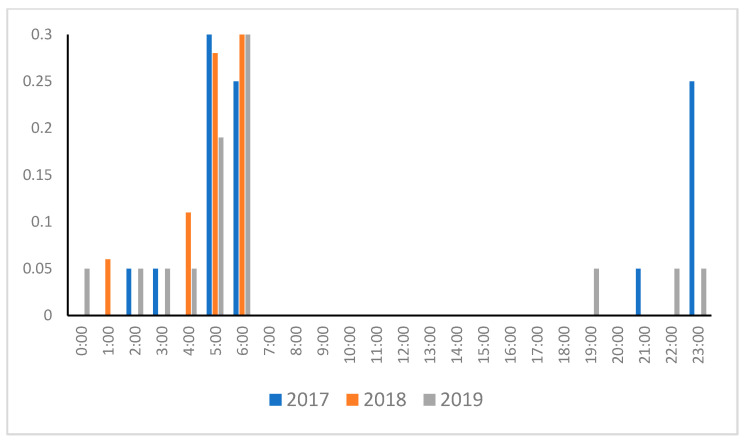
Water entry start time from January to March 2017–2019.

**Figure 4 animals-14-01312-f004:**
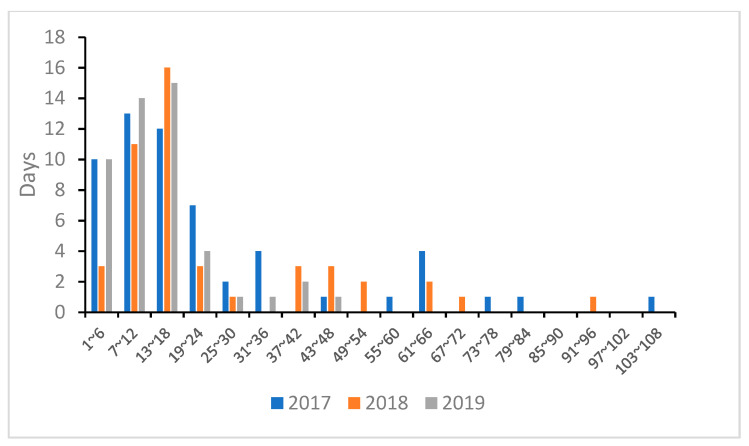
Distribution of residence time from January to March in 2017–2019.

**Table 1 animals-14-01312-t001:** The herring catch and the highest number of SSLs in a day.

	2017		2018		2019	
	Herring	Max	Herring	Max	Herring	Max
Early Jan	0	356	0	464	0	253
Mid-Jan	3	372	212	615	9	232
Late Jan	59	702	143	524	53	362
Early Feb	139	530	286	357	203	67
Mid-Feb	158	535	119	276	213	129
Late Feb	279	572	238	131	114	18
Early Mar	70	381	95	47	185	13

Herring: herring catch (ton); max: highest number of SSLs on the haulout each day; Jan: January; Feb: February; Mar: March; early: 1st to 10th day; mid: 11 to 20th day; late: 21st to the last day of the month.

## Data Availability

The data presented in this study are not publicly available. Please contact the corresponding author with any inquiries.

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
