# Peer review of "Human Impact on the Twenty-Four-Hour Patterns of Steller Sea Lions’ Use of a Haulout in Hokkaido, Japan"

_animals, 2024, doi:10.3390/ani14091312_

Round 1
Reviewer 1 Report
Comments and Suggestions for Authors
The authors describe haulout use by Steller Sea Lions in response to weather and human disturbance. They clearly demonstrated SSL use of a haulout but did a poor job in describing human activities near the haulout and how the SSL impacted the herring fishery. There was apparent extermination of SSL on the haulout which was not adequately described. Overall I don’t think the authors did a good job describing how their method supported their conclusions.
Specific lines that were confusing:
Lines 146-147-“If SSLs learn through repeated shooting each season, neither the number of SSLs using the haulout nor the re- maining time on the haulout will decrease” I don’t follow this logic.
Lines 177-178-“Extermination of SSLs was conducted seven times at the haulout starting in late February. Even after the extermination began, SSLs came ashore, but their numbers decreased in March” please clarify what you mean by extermination
Line 261-“As a result, the remaining time on the haulout peaked at 7-12 hours in 2017, but the longest time was 104 hours on March 3-4.” This doesn’t make sense how can 104 hours occur over 2 days?
Lines 303-304-“This suggests that SSLs remember instances of being hunted when people shot rifles from boats approaching from the front of the haulouts” this hunting should have been described earlier in the paper. Perhaps this is the extermination that was refereed to earlier?
Line 383-“Data over the past ten years show that fishery damages caused by SSLs in Ishikari Bay have remained almost unchanged” the damage on the herring fishery by SSL is not clearly described in the paper and leaves the reader unsure about the SSL impact on the fishery.
Comments on the Quality of English LanguageEnglish was relatively clear
Author Response
Dear Sir or Madam in Reviewer 1
We appreciate your taking the time to review our manuscript. We modified it according to your comments, and those sentences were highlighted. We also added some more information about the history and management of SSL in the Introduction. We hope those modifications meet your comments.
Specific lines that were confusing:
Lines 146-147-“If SSLs learn through repeated shooting each season, neither the number of SSLs using the haulout nor the remaining time on the haulout will decrease” I don’t follow this logic.
>The hunters shoot SSLs mainly at this haulout where SSLs were on. SSLs have become sensitive to any boat approach and other human disquieting movements. Furthermore, they also learned when the hunter started hunting. So, they couldn’t have rest as long as they wanted. These SSI’s psychological unbalance probably induced the shortage of haulout using time.
Lines 177-178-“Extermination of SSLs was conducted seven times at the haulout starting in late February. Even after the extermination began, SSLs came ashore, but their numbers decreased in March” please clarify what you mean by extermination
>We also rethink that “extermination” makes readers imagine killing every SSL on the haulout. So, we changed the word “extermination” to “hunting” or “culling”.
Line 261-“As a result, the remaining time on the haulout peaked at 7-12 hours in 2017, but the longest time was 104 hours on March 3-4.” This doesn’t make sense how can 104 hours occur over 2 days?
>This number did not always show that the same individuals or groups remained but showed any number of SSLs on the haulout.
Lines 303-304-“This suggests that SSLs remember instances of being hunted when people shot rifles from boats approaching from the front of the haulouts” this hunting should have been described earlier in the paper. Perhaps this is the extermination that was refereed to earlier?
>Thank your for your advice. We modified and added this sentence in the Introduction L124-125.
Line 383-“Data over the past ten years show that fishery damages caused by SSLs in Ishikari Bay have remained almost unchanged” the damage on the herring fishery by SSL is not clearly described in the paper and leaves the reader unsure about the SSL impact on the fishery.
>We added a Supplemental table about the 10-year fishery damage by SSLs. We hope that the table covers that sentence.
We appreciate your time.
Best regards
Yuko CHAYAGHARA, Yumiko NAKANOWATARU, and Takanori KOORIYAMA,
Department of Veterinary Science,
School of Veterinary Medicine,
Rakuno Gakuen University
Reviewer 2 Report
Comments and Suggestions for Authors
I read carefully and with great expectations the paper "Human Impact on the Twenty-Four-Hour Patterns of Steller Sea Lions' Use of Haulout in Hokkaido, Japan". However, I did not find, overall, the theme to be well developed nor suited to the quality of the "Animal" journal. The data is analyzed in a very general way and a suitable in-depth framework is not offered. More statistical analyzes would be needed as well as a better structuring of the discussion and conclusions. Furthermore, I find no evidence that Steller sea lions are responsible for the decline in herring. Dynamics like these probably involve many factors, often also due to human activity (overfishing for example) or environmental and climate changes. In general I believe that overall the paper is not suitable for publication.
Author Response
Dear Sir or Madam in Reviewer 2
Thank you for reviewing our manuscript.
We agree that our manuscript needed modification and clarification for research purposes. We had difficulties combining topics such as SSL-fishery conflict and SSL behavior change by human disturbance. However, this issue, conflicts between SSL management and fishery damage, is a sensitive topic in Japan because public opinion tends to be on the side of fishermen who suffered damage from SSLs. Even researchers find it difficult to state their opinion on conservation for fear of being accused. So, our study aims to report the human impact on the SSLs without inducing emotional responses.
In the modified manuscript, we added the Japanese history of SSL management, its commercial use, and public opinion of SSL fishery damage in Japan. The analysis was performed simply because several factors affect hauling out and diving into the water. So, we ignored the indirect factors and focused on more direct factors.
We modified and added several sentences in the manuscript, as highlighted. We hope we have the comments on the renewed manuscript.
We appreciate your time.
Best regards
Yuko CHAYAGHARA, Yumiko NAKANOWATARU, and Takanori KOORIYAMA,
Department of Veterinary Science,
School of Veterinary Medicine,
Rakuno Gakuen University
Reviewer 3 Report
Comments and Suggestions for Authors
Dear Authors,
thanks for your invitation to review your interesting work.
I'm afraid you would need to give another perspective to your work starting from your introduction.
You are presenting marine mammals as competitors, damage-makers, etc. without any mention of their primary ecological role in marine ecosystem. The concept at the basis of your work is anthropocentric and no mention of the importance of conservation is provided with the correct approach.
Protecting marine mammals was a planet's goal and not an issue for human resources. Marine mammals and human interaction is a social and ecological problem that can be solved with mitigation tools and not by eliminating these fundamental animals for the ecosystem.
Please, provide a better introduction of your work reading and using better and more recent references, also from other countries and realities.
The goals of the work are quite clear but there is not a precise aspect investigated: why you are studying the conflict? To protect the seals or to protect the work of the fishermen?
It's a bit anachronistic not to mention a conservation aspect in a work studying marine animals in nature. Please, try to go deeper into this topic and improve your introduction and discussion based on your method.
Many thanks
Author Response
Dear Sir or Madam of reviewer 3
Thank you very much for your time reviewing our manuscript. We noticed that this manuscript needed more brush up. We modified it with your helpful comments. We hope the correction meets your suggestion.
I'm afraid you would need to give another perspective to your work starting from your introduction.
You are presenting marine mammals as competitors, damage-makers, etc. without any mention of their primary ecological role in marine ecosystem. The concept at the basis of your work is anthropocentric and no mention of the importance of conservation is provided with the correct approach.
>We inserted the importance of marine mammal migration to the coast in the Conclusion in Lines 427-431. We will relocate and modify the sentences if we should note this information in the Introduction.
Protecting marine mammals was a planet's goal and not an issue for human resources. Marine mammals and human interaction is a social and ecological problem that can be solved with mitigation tools and not by eliminating these fundamental animals for the ecosystem.
Please, provide a better introduction of your work reading and using better and more recent references, also from other countries and realities.
>We added the history of SSL management in Japan and the SSL decline in recent years in the Introduction, Lines 52-63. We also added information about the history of SSL management in Japan and the decline of the SSL population around Japan and Russia in the Introduction in Lines 77-93.
The goals of the work are quite clear but there is not a precise aspect investigated: why you are studying the conflict? To protect the seals or to protect the work of the fishermen?
It's a bit anachronistic not to mention a conservation aspect in a work studying marine animals in nature. Please, try to go deeper into this topic and improve your introduction and discussion based on your method.
>Our manuscript lacks a view of marine mammal conservation. So, we added what the researcher should recommend to the fishermen and public in Lines 431-435.
We appreciate your time.
Best regards
Yuko CHAYAGHARA, Yumiko NAKANOWATARU, and Takanori KOORIYAMA,
Department of Veterinary Science,
School of Veterinary Medicine,
Rakuno Gakuen University
Reviewer 4 Report
Comments and Suggestions for Authors
This article covers a 'classic' theme in human-wildlife conflict, namely potential or real impacts of an aquatic piscivorous animal on fishery interests. It covers a relatively long time period, three winter seasons, but is focused on a single site. Constant video monitoring means there was no need for sampling but clearly a lot of effort was required in processing images.
In the introduction, I would like to see the authors give a fuller account of the history of shooting/control of SSL at their study site and maybe put this in the context of other piscivore/fishery conflicts elsewhere in Japan and how they are managed. Is the scaring/disturbance/control of SSL on Hokkaido legal and is there any opposition to it from wildlife conservation groups? Where does the Otaru Aquarium stand on this issue: neutral, pro-wildlife or pro-fishery?
Some generic points on alternative wording are given in the box on improving the English language, but other comments are given here that may improve the overall readability.
Title: insert 'a' between 'of' and 'haulout'
Line 16: replace beach with coast
Line 25: insert 'interests' at end of sentence
Line 37: suggest 'damage caused to the herring fishery'
Line 45-46: replace fishery resources with fish stocks
Line 62: replace 'limited places' with 'a few locations'
Line 68: replace a 'massive blow' with 'at a considerable cost' to the herring fishery
Line71-72: the upper limit changed to 116 - from what?
Line 75: replace 'token' with 'quota'
Line 78: suggest 'meat circulates'
Line 83-84: some repetition here so delete 'can land. Over one hundred SSLs'
Line 85: replace 'theorised' with 'hypothesised'
Line 90: was this (now retired) hunter actively preventing the use of the haulout prior to the present study?
Line 91: replace 'extermination' with 'shooting'.
Line 95: replace 'flocks' with 'herds'
Line 98: replace 'fixing' with 'resolving'
Line 105: some more repetition here, delete ' off the coast of Shukutsu, Otaru City, Hokkaido' inset 'at' (and remove parentheses around the coordinates)
Line 114: simply state 'in each year from 2017 to 2019'
Line 120: what is a warning boat and define their action please. Do you mean 'non-lethal scaring'?
Line 142: replace 'sensitive to' with 'wary of'
Line 147: replace 'neither' with 'either' and 'nor' with 'and'
Line 155: delete 'by' and 'analysed'
Line 178: replace 'extermination' with 'control efforts'
Line 191: replace 'droves' with 'significant numbers'
Line 282/x-axis should read 'hours' not 'hous'
Line 297: replace 'attacked by solid' with 'faced with serious'
Line 299: replace 'swallow' with 'overwash'
Line 310: replace 'famous' with 'renowned'
Line 315: replace 'abundant' with 'abundance of'
Line 334: suggest 'on a breakwater'
Line 336: replace 'escape from' with 'desert'
Line 337: insert 'the' after 'still'
Line 338: insert 'new' after 'this'
Line 341: replace 'broad' with 'catholic' and 'picky' with 'selective'
Line 355: replace 'not very affected' with 'virtually unaffected'
Line 360: reword 'from loudspeakers' and please explain what 'bear bunkers' are
Line 369: male SSLs fast while maintaining their territory
Line 384: perhaps replace 'eradicating' with 'controlling'
Line 391: suggest 'a large total catch may encourage fishermen to overlook' the damage.....
Line 393: maybe 'fishery resources/stocks'
Comments on the Quality of English Language
Widespread use of 'extermination' (lines 48, 69, 71 etc.) is not appropriate. Perhaps this should be replaced with 'lethal control' or 'culling'. Extermination implies total removal of a local population rather than selective removal of individuals or regular disturbance to 'move the SSL elsewhere'.
Remember 'haulout' can be used as both a verb (to haulout) and a noun (a site such as Todo Iwa)
Perhaps consider replacing 'remaining time' with 'residence time' (line 145 onwards)
Author Response
Dear Sir or Madam of Reviewer 4
Thank you for your kind comments and advice.
We tried to combine two points, SSL conservation and behavior changes by human impact, in this manuscript, but we had some difficulties binding them. Your comments and corrections greatly helped us modify this manuscript. We appreciate your review. The modified part of the sentences was highlighted. We hope our renewed manuscript meets your review.
In the introduction, I would like to see the authors give a fuller account of the history of shooting/control of SSL at their study site and maybe put this in the context of other piscivore/fishery conflicts elsewhere in Japan and how they are managed.
> We added some information about management to four pinnipeds and also the history of SSL management in Japan in the introduction in Line 52-63 and Line 77-93.
Is the scaring/disturbance/control of SSL on Hokkaido legal and is there any opposition to it from wildlife conservation groups? Where does the Otaru Aquarium stand on this issue: neutral, pro-wildlife or pro-fishery?
> Not only Otaru aquariums but also other aquariums in Japan rarely object to the lethal control of SSLs and other marine mammals; one of their aims is to protect the conservation of endangered animals. One reason is that they buy fish to feed animals from fishermen. So, they worried about being denied to sell fishes by fishermen. The other is that public opinion is concerned with fishery damage rather than marine mammal conservation. We added the kind of this topic in Line 104-107.
Some generic points on alternative wording are given in the box on improving the English language, but other comments are given here that may improve the overall readability.
> Your comments are a great help to us. We kindly adopt your advice.
Title: insert 'a' between 'of' and 'haulout'
> Yes, we did.
Line 16: replace beach with coast
> Yes, we did.
Line 25: insert 'interests' at end of sentence
> Yes, we did.
Line 37: suggest 'damage caused to the herring fishery'
> We adopted your suggestion.
Line 45-46: replace fishery resources with fish stocks
> Yes, we did.
Line 62: replace 'limited places' with 'a few locations'
> Yes, we did.
Line 68: replace a 'massive blow' with 'at a considerable cost' to the herring fishery
> Yes, we did.
Line71-72: the upper limit changed to 116 - from what?
> This meant “from no limit.” So, we inserted “unrestricted” after “…to 116”
Line 75: replace 'token' with 'quota'
> Yes, we did.
Line 78: suggest 'meat circulates'
> We agree with this suggestion.
Line 83-84: some repetition here so delete 'can land. Over one hundred SSLs'
> We modified this sentences.
Line 85: replace 'theorised' with 'hypothesized'
> Yes, we did.
Line 90: was this (now retired) hunter actively preventing the use of the haulout prior to the present study?
> Yes. The retired hunter shot SSLs that resting on the haulout first but he also aim at the sea surface.
Line 91: replace 'extermination' with 'shooting'.
> Yes, we did.
Line 95: replace 'flocks' with 'herds'
> Yes, we did.
Line 98: replace 'fixing' with 'resolving'
> Yes, we did.
Line 105: some more repetition here, delete ' off the coast of Shukutsu, Otaru City, Hokkaido' inset 'at' (and remove parentheses around the coordinates)
>We also modified these.
Line 114: simply state 'in each year from 2017 to 2019'
>We also modified it.
Line 120: what is a warning boat and define their action please. Do you mean 'non-lethal scaring'?
>Yes, it is. I would use your words. It’s helpful.
Line 142: replace 'sensitive to' with 'wary of'
> Yes, we did.
Line 147: replace 'neither' with 'either' and 'nor' with 'and'
> Yes, we did.
Line 155: delete 'by' and 'analysed'
> Yes, we did.
Line 178: replace 'extermination' with 'control efforts'
> Yes, we did.
Line 191: replace 'droves' with 'significant numbers'
> Yes, we did.
Line 282/x-axis should read 'hours' not 'hous'
> We corrected it.
Line 297: replace 'attacked by solid' with 'faced with serious'
> Yes, we did.
Line 299: replace 'swallow' with 'overwash'
> Yes, we did.
Line 310: replace 'famous' with 'renowned'
> Yes, we did.
Line 315: replace 'abundant' with 'abundance of'
> Yes, we did.
Line 334: suggest 'on a breakwater'
> We agree with this suggestion.
Line 336: replace 'escape from' with 'desert'
> Yes, we did.
Line 337: insert 'the' after 'still'
> Yes, we did.
Line 338: insert 'new' after 'this'
> Yes, we did.
Line 341: replace 'broad' with 'catholic' and 'picky' with 'selective'
> Yes, we did.
Line 355: replace 'not very affected' with 'virtually unaffected'
> Yes, we did.
Line 360: reword 'from loudspeakers' and please explain what 'bear bunkers' are
> We modified it as “making loud noises from the speaker.” And we added the explanation (explosion sound) of 'bear bunkers' after it.
Line 369: male SSLs fast while maintaining their territory
> We corrected this.
Line 384: perhaps replace 'eradicating' with 'controlling'
> We agree with this suggestion.
Line 391: suggest 'a large total catch may encourage fishermen to overlook' the damage.....
> We agree with this suggestion.
Line 393: maybe 'fishery resources/stocks'
> We agree with this suggestion.
Comments on the Quality of English Language
Widespread use of 'extermination' (lines 48, 69, 71 etc.) is not appropriate. Perhaps this should be replaced with 'lethal control' or 'culling'. Extermination implies total removal of a local population rather than selective removal of individuals or regular disturbance to 'move the SSL elsewhere'.
>Thank you for your kind advice. We had confused the meanings of these words after English editing services. We adopted your advice and changed the words.
Remember 'haulout' can be used as both a verb (to haulout) and a noun (a site such as Todo Iwa)
>Yes. We had hesitated to use haulout as a verb but used land. We replace “land” with “haulout.”
Perhaps consider replacing 'remaining time' with 'residence time' (line 145 onwards)
>Your suggestion is also helpful for us. We felt that something wasn’t quite right.
We appreciate your time.
Best regards
Yuko CHAYAGHARA, Yumiko NAKANOWATARU, and Takanori KOORIYAMA,
Department of Veterinary Science,
School of Veterinary Medicine,
Rakuno Gakuen University
Round 2
Reviewer 2 Report
Comments and Suggestions for Authors
The authors have improved the quality of the paper. They answered the questions posed. Therefore I believe that the paper can now be published
Reviewer 3 Report
Comments and Suggestions for Authors
Dear authors, thanks for improving the manuscript which I found very interesting and in line with the current global conservation goals for this species.